# Nurturing the Will: Unraveling Associative Factors of Exclusive Breastfeeding Intentions Among Primigravida in Saudi Arabia

**DOI:** 10.3390/healthcare13222938

**Published:** 2025-11-17

**Authors:** Asmaa Mohamed Ali AlAbd, Aziza Ibrahim Mohamed Hassan, Salwa Ali Marzouk, Mahmoud Abdelwahab Khedr, Jawza Hmeid Hmood Alrashidi, Maha Sanat Alreshidi, Abdulhafith Alharbi, Sarah Sulieman Al Nawasreh

**Affiliations:** 1Psychiatric and Mental Health Nursing Department, College of Nursing, University of Hail, Hail 81481, Saudi Arabia; a.alabd@uoh.edu.sa (A.M.A.A.); ay.alharbi@uoh.edu.sa (A.A.); 2Department of Nursing, College of Applied Medical Science, University of Bisha, Bisha 61922, Saudi Arabia; aimhassan@ub.edu.sa; 3Maternal and Child Health Nursing Department, College of Nursing, University of Hail, Hail 81481, Saudi Arabia; s.marzouk@uoh.edu.sa; 4College of Nursing, University of Hafr Albatin, Hafr Albatin 39511, Saudi Arabia; 5Department of Psychiatric and Mental Health Nursing, Faculty of Nursing, Alexandria University, Alexandria 21511, Egypt; 6Community Health Nursing Department, College of Nursing, University of Hail, Hail 81481, Saudi Arabia; j.alreshedy@uoh.edu.sa; 7Administration Department, College of Nursing, University of Hail, Hail 81481, Saudi Arabia; ma.alrashidi@uoh.edu.sa; 8College of Nursing, University of Hail, Hail 81481, Saudi Arabia; s.alnawasrah@uoh.edu.sa

**Keywords:** associative factors, exclusive breastfeeding intentions, primigravida

## Abstract

**Background:** Exclusive breastfeeding is widely acknowledged as beneficial for a baby’s health and development. However, Saudi Arabia remains one of the few countries where this practice is still uncommon. This study explores the factors associated with exclusive breastfeeding intention among primigravida women in Saudi Arabia. **Methods**: A cross-sectional design and systematic random sampling technique were utilized to collect data from a total of 436 primigravida women. **Results**: More than half of the participants (58.3%) expressed high intentions toward exclusive breastfeeding. The most significant modifiable predictors of exclusive breastfeeding intention among these women were age, education, nationality, and planned pregnancy. Additionally, adequate knowledge of breastfeeding (Adjusted Odds Ratio [AOR]: 0.402, Confidence Interval [CI]: 0.226–0.717) and a positive attitude toward breastfeeding (AOR: 0.396, CI: 0.223–0.705) emerged as independent predictors of high breastfeeding intention. **Conclusions**: The study revealed that while more than half of the primigravida women demonstrated a high intention to practice exclusive breastfeeding (EBF), this level of intention is still considered inadequate for achieving optimal breastfeeding practices. Key factors influencing their breastfeeding attitudes included maternal age, education level, ethnicity, pregnancy planning, and knowledge about breastfeeding.

## 1. Background

According to the World Health Organization (WHO), exclusive breastfeeding (EBF) during the first six months of an infant’s life is essential for the health and well-being of both mothers and babies, providing numerous nutritional and immunological benefits [1]. Despite the presence of cultural and non-religious customs in Saudi Arabia that support breastfeeding, the rate of exclusive breastfeeding in the country remains below the international average [2]. This gap between intention and actual practice, particularly among primigravida (first-time mothers), poses significant public health challenges.

Breastfeeding offers substantial benefits for both mothers and their infants. It is an energy-intensive process that helps mothers return to their pre-pregnancy weight more effectively [3]. Similarly, exclusive breastfeeding is linked to a lower incidence of breast cancer, longer periods of lactation amenorrhea, and the maintenance of a healthy weight [4]. Breastfeeding can reduce newborn mortality by 13% and decrease the incidence of sudden infant death syndrome by 74% [5].

Primigravida mothers encounter unique challenges, including a loss of enjoyment, anxiety about breastfeeding techniques, and misconceptions regarding the advantages of breastfeeding. These factors can significantly hinder their ability to initiate and sustain exclusive breastfeeding. Previous studies indicate that key determinants of a mother’s intention to breastfeed include parental confidence, social support, access to healthcare guidance, and prenatal education [6,7]. In Saudi Arabia, cultural norms, misinformation, and a lack of qualified breastfeeding support in both hospital and community settings further complicate the breastfeeding experience for first-time mothers [8].

Additionally, the preferences of primigravida women regarding exclusive breastfeeding are heavily influenced by their knowledge levels. Historically, many women have entered pregnancy with limited or inaccurate information about breastfeeding practices. A study revealed that while most Saudi mothers understood the benefits of breastfeeding, they lacked awareness about the significance of exclusive breastfeeding for the first six months, the nutritional value of breast milk, and its role in enhancing infant immunity [9]. This knowledge gap is concerning, as it negatively impacts both the duration and exclusivity of breastfeeding.

A mother’s attitude toward breastfeeding plays a critical role in shaping her intentions and ability to navigate various breastfeeding scenarios. Positive beliefs—such as the understanding that breastfeeding promotes infant health and strengthens maternal bonding—are associated with higher rates of initiation and continuation. Conversely, negative attitudes or doubts, including fears of pain, concerns about insufficient milk supply, or the perception that formula feeding is more convenient, can diminish the likelihood of establishing early breastfeeding [10].

Cultural norms and societal expectations significantly influence attitudes toward breastfeeding in Saudi Arabia. While family traditions and religious teachings generally advocate for breastfeeding as a vital practice, opinions are evolving, particularly among younger, first-time mothers. Contemporary lifestyle changes, urbanization, and the increasing influence of advertisements for infant formula as a breast milk substitute have contributed to this transition. Primigravida women may face conflicting messages; although traditional practices endorse breastfeeding, cultural constraints and a lack of support can make breastfeeding challenging [7].

Understanding the factors that influence primigravida women’s preferences for breastfeeding is crucial for developing interventions that can bridge the gap between intention and sustained breastfeeding practice. This study aims to contribute to the existing body of literature by providing an in-depth examination of the cultural, psychological, and scientific factors unique to Saudi primigravida women. Consequently, our research seeks to explore the intentions of Saudi Arabian primigravida women regarding exclusive breastfeeding and identify the variables associated with these intentions.

## 2. Methods

Research design: This study utilized a cross-sectional research methodology to investigate the intention of primigravida women to exclusively breastfeed and the factors influencing this intention.

Operational Definition: For this study, “exclusive breastfeeding” (EBF) is defined as feeding an infant only breast milk for the first six months of life, with no additional food or drink, including water, except for oral rehydration solutions, drops, or syrups containing vitamins, minerals, or medicines, as specified by the World Health Organization [11,12].

Setting: The research was conducted in the prenatal follow-up clinics at Maternal and Child Hospitals (MCH) in the Saudi Arabian cities of Hail and Bisha. MCH specializes in maternal and child health services, providing obstetric care to a substantial segment of the local population. As a wealthy nation, Saudi Arabia’s Ministry of Health prioritizes the overall health of mothers and children, actively promoting knowledge and practices related to exclusive breastfeeding [13].

Participants: The research sample was obtained using a systematic random sampling technique. This involved establishing a sampling interval (k) by dividing the total number of eligible first-time pregnant women attending the follow-up clinic for antenatal clinic services by the desired sample size. A random starting point was selected from 1 to k, and every k-th eligible woman on the clinic list was recruited until the target sample size was achieved. Inclusion Criteria: Participants in the study were required to be women in their third trimester of pregnancy, aged 18 years or older. To ensure the health and safety of the participants and their infants, women with significant health issues were excluded from the study. Significant health issues were determined based on self-reported medical history and included any chronic conditions or severe illnesses that could impair the ability to breastfeed or pose risks during pregnancy, including mental illnesses. Additionally, women who were multigravida, those who declined to participate, or those with contraindications for breastfeeding—such as HIV, hepatitis B, or a history of previous breast surgeries—were also excluded from the study.

Sample Size Calculation: The sample size was determined using the formula for a single population proportion:$$n=\frac{(Z1-\frac{\alpha }{2})2P(1-P)}{d2}$$

In this equation, (P) represents the estimated percentage of mothers with a high intention to breastfeed (P = 56.3%) based on a previous study [14]. Here, (n) denotes the sample size, (d) is the margin of error (0.05), and (Z1 − α/2) is the standard normal variable at a 95% confidence level (1.96). After accounting for a 15% non-response rate, the final calculated sample size was 436 participants.

Study Tools:

The researchers developed a self-administered questionnaire to collect data, which included four sections:

The first section collected sociodemographic and general characteristics of the participants, including age, education levels, residency, occupation, family structure, sources of information about breastfeeding, and reproductive history.

The second section focused on examining the participants’ knowledge regarding Exclusive Breastfeeding (EBF), which was adapted from Dukuzumuremyi et al. [15]. This section contained twenty questions assessing pregnant women’s understanding of exclusive breastfeeding. Items included the significance of EBF, duration of breastfeeding, early initiation, the right timing for introducing complementary foods, the importance of colostrum, benefits for mothers and infants, and risks associated with bottle-feeding. Content validity was established through a review of current and historical literature relevant to breastfeeding practices, including WHO guidelines [12,15]. The scoring method assigned a “1” for a correct response and “0” for an incorrect response, resulting in a total score ranging from 0 to 20. Participants were categorized as having adequate knowledge if they scored 12 or higher (≥60%) and inadequate knowledge if they scored below 12 (<60%). These cutoffs were selected based on prior studies indicating that scores of 60% or above reflect a sufficient understanding of breastfeeding practices necessary for promoting EBF [14,16].

The third section comprised questions to assess Attitudes toward Exclusive Breastfeeding (EBF): This section, also adapted from Dukuzumuremyi et al. [15], included 21 questions designed to measure women’s attitudes toward exclusive breastfeeding. Topics covered: perceptions of colostrum, introducing complementary foods before six months is essential, the sufficiency of EBF for up to six months, the convenience of formula feeding, the health benefits of EBF, the bonding experience associated with breastfeeding, the superiority of EBF over formula, etc. The content validity was ensured by reviewing existing literature, including cultural considerations. Responses were measured on a five-point Likert scale (from strongly agree to strongly disagree), yielding scores ranging from 1 to 105. A positive attitude was defined as a score between 63 and 105, while scores between 1 and 62 indicated a negative attitude. The choice of these cutoffs was based on established norms from similar studies, which suggest that a score above 60% of the maximum indicates a generally favorable attitude toward breastfeeding [15].

The fourth section: Exclusive Breastfeeding Intentions (EBFI) Scale: Adapted from Yeha et al. [17], this scale assessed maternal intentions to breastfeed exclusively. The EBFI questionnaire is a five-item scale rated on a five-point Likert scale, where a score of 1 indicated strong disagreement and 5 indicated strong agreement. Participants scoring above the median intention scale of 13 were classified as having a firm breastfeeding intention, while those scoring below were identified as having a low intention. The scale demonstrated good internal consistency, with a Cronbach’s alpha of 0.82, confirming its reliability in the Saudi context [17].

To ensure cultural relevance, the questionnaire items were translated into Arabic according to the World Health Organization’s recommendations for instrument adaptation [12]. The translated versions underwent a critical review by qualified academics and medical professionals specializing in maternal and pediatric nursing, focusing on face and content validation and the use of appropriate terminology for Arabic speakers [18].

Data Collection Methods:

A questionnaire was initially developed in English and subsequently translated into Arabic. The translation process involved a bilingual expert who performed a forward translation from English to Arabic, followed by a backward translation into English to ensure accuracy. The back-translated version was then compared with the original English version for consistency. To validate the questionnaire, five professors specializing in Public Health and Maternal and Child Health reviewed it before data collection, and necessary modifications were made based on their feedback.

Following the development of the tool, a pilot study was conducted involving 40 pregnant women from each Maternal and Children Hospital (MCH) in Hail and Bisha Cities to assess the relevance and comprehensibility of the study tools, as well as to determine the time required for completion; these participants were excluded from the main study. The internal consistency of the tool was measured using Cronbach’s alpha, yielding a reliability coefficient of R = 0.87.

Data collection occurred between late April and late August 2024, with the data collection team operating three days a week until the desired sample size was achieved. Utilizing systematic random sampling, the clinic nurses reported an average of 60 cases per day across the four prenatal clinics. The data collection team, consisting of two researchers, aimed to collect data from 10 participants daily. The sampling interval was calculated as 60/10 = 6. A random starting point was chosen from 1 to 6, and the sampling interval was applied accordingly.

An Arabic-language questionnaire was used for data collection. To prevent duplicate data collection, each mother’s clinic card was marked with an asterisk before the data collection commenced. Confidentiality was maintained throughout the interview sessions to ensure participants felt comfortable completing the questionnaire. Before the interview, participants were informed about the survey’s topic and format, with assurances that their responses would remain confidential. Written informed consent was obtained at the beginning of the questionnaire, and participants were made aware that they could withdraw from the study at any time without any repercussions or compensation.

Data Processing and Analysis:

Data collection was facilitated using EpiData version 3.1, which was subsequently exported to SPSS version 25 for comprehensive analysis. We employed descriptive statistics to outline the sociodemographic characteristics of participants, alongside their knowledge and attitudes regarding breastfeeding. A multivariable logistic regression analysis was done to assess the effect of various study factors on breastfeeding intentions, and the results were tabulated as adjusted odds ratios (AORs) and confidence intervals (CIS). *p*-value (<0.05) was adopted as the level of significance.

## 3. Results

The study comprised 436 primigravida women, with the majority (73.6%) identifying as Saudi nationals. A significant portion of our participants (50%) were aged between 25 and 30 years. Data collection occurred across various healthcare facilities, yielding nearly equal representation across our sampling locations (50.5% from one site and 49.5% from another). Most participants (64.4%) held university degrees, and over three-quarters (78.7%) were unemployed. Notably, around 76.1% reported moderate income levels, as illustrated in Table 1.

Table 2 presents the distribution of obstetric history among the study participants. A significant majority of the primigravida women (81.9%) were carrying a single fetus, while 70.4% indicated that they had not experienced any previous abortions. Furthermore, most participants (86.9%) reported no complications during their pregnancies and maintained regular follow-up appointments (67.4%). Interestingly, more than one-third of the women (35.8%) noted that their pregnancies were unplanned.

Table 3 highlights participants’ knowledge, attitudes, and intentions regarding exclusive breastfeeding. Over half of the participants displayed adequate knowledge (54.4%), a positive attitude (53.7%), and a high intention to pursue exclusive breastfeeding (58.3%).

A multivariable logistic regression analysis presented in Table 4, several sociodemographic and obstetric factors are independently associated with high intention to exclusively breastfeed among participants. Notably, age played a crucial role; participants aged 30 years and older were 3.26 times more likely to have a high intention compared to those aged 20 to less than 25 years (AOR = 3.26, 95% CI: 1.24–8.60, *p* = 0.017). Education level also proved significant, as participants with postgraduate degrees showed over six times higher odds of high intention than those with secondary education (AOR = 6.19, 95% CI: 1.63–23.56, *p* = 0.008). In contrast, high-income participants were less likely to report high intention compared to those with low income (AOR = 0.16, 95% CI: 0.04–0.64, *p* = 0.010). Additionally, non-Saudi participants had significantly higher odds of high intention (AOR = 2.47, 95% CI: 1.20–5.09, *p* = 0.014), while those with unplanned pregnancies had 59% lower odds of high intention (AOR = 0.41, 95% CI: 0.22–0.76, *p* = 0.005). Furthermore, participants demonstrating adequate knowledge (AOR = 0.40, 95% CI: 0.23–0.72, *p* = 0.002) and a positive attitude (AOR = 0.40, 95% CI: 0.22–0.71, *p* = 0.002) were significantly more likely to report high intention. Overall, these findings indicate that older age, higher educational level, non-Saudi nationality, adequate knowledge, and a positive attitude are key predictors of high intention to breastfeed, while high income and unplanned pregnancies are associated with reduced intention levels.

## 4. Discussion

Breastfeeding is universally recognized as the essential and exclusive source of nutrition for infants up to six months of age, playing a crucial role in their growth and development. However, among Saudi women, the rates of exclusive breastfeeding (EBF) during the first six months fall significantly short of the World Health Organization (WHO) recommendations, which advocate for exclusive breastfeeding rates of at least 50% [12]. This study highlights the importance of understanding pregnant women’s intentions regarding exclusive breastfeeding early in their pregnancy, as these intentions are pivotal in ensuring higher initiation rates of EBF.

One of the key factors influencing actual breastfeeding behavior is intention. In our study, we found that over half of the first-time mothers (primigravida women) expressed strong intentions to breastfeed exclusively. However, this is somewhat disappointing because it doesn’t quite meet the World Health Organization (WHO) targets, which recommend that at least 50% of infants be exclusively breastfed during their first six months [12]. Interestingly, this finding is consistent with the research by Ibrahim et al. [14], where they reported that 56.3% of participants had high intentions for breastfeed exclusively. Furthermore, approximately 50% of respondents said they would exclusively breastfeed their infants, according to a cross-sectional study conducted at King Saud University Medical Center to assess the impact of waiting for mothers’ expertise and attitude toward the goal of doing so [19]. Yet, previous studies have shown that there are regions in Saudi Arabia with higher breastfeeding initiation rates [16,20].

Despite the presence of breastfeeding intentions among the participants, actualizing these intentions requires a multifaceted approach that addresses various influencing factors. In our analysis, several independent predictors of high breastfeeding intentions emerged, including participants’ knowledge and attitudes towards breastfeeding, maternal age, educational background, nationality, and pregnancy planning.

The findings revealed that only 54.4% of participants demonstrated adequate knowledge regarding exclusive breastfeeding, indicating a significant gap in public awareness and suggesting that educational initiatives need to be intensified in Saudi Arabia. This result is comparable to international studies, such as those conducted in Calabar, Nigeria (56.8%) and Northwest Ethiopia (57.8%) [21,22]. A descriptive cross-sectional study involving 382 pregnant women in Saudi Arabia between November 2022 and January 2023 found that more than half of the participants knew enough about EBF [14]. In contrast, Elgzar et al. [23] reported that over two-thirds of lactating women in Najran had sufficient knowledge about EBF, highlighting a discrepancy that may arise from different study populations. Primigravida women often face unique challenges, including a lack of experience and potential misconceptions about breastfeeding benefits, which can hinder their ability to initiate and maintain EBF.

In addition, the current study showed that maternal knowledge regarding EBF is one of the significant predictors of high breastfeeding intentions. It would seem reasonable that sufficient knowledge and effective EBF procedures are related; a woman should have enough knowledge to adopt any behavior. Therefore, it is important to emphasize educational programs to promote breastfeeding. Supporting this view, several studies in Saudi Arabia have stated that after participating in educational sessions, EBF procedures significantly improved [24,25]. These results emphasize the significance of focused educational initiatives to improve maternal awareness and support breastfeeding.

Regarding attitudes towards exclusive breastfeeding, the study found that 53.7% of participants held a positive attitude. This is influenced by social norms and cultural beliefs, particularly in Saudi Arabia, where religious convictions play a vital role in breastfeeding practices [14]. The teachings of the Holy Qur’an, which encourage breastfeeding for up to two years, further support this positive attitude [26]. In this regard, a positive correlation between cultural beliefs and breastfeeding attitudes in Saudi women [27].

The theory of planned behavior suggests that a mother’s attitude, along with her beliefs and perceived control over her actions, plays a crucial role in shaping her breastfeeding intentions and behaviors. Many previous studies have shown a strong link between attitude and intention in various health-related behaviors [14,15,16,28]. Similarly, our study found a significant relationship between attitudes and intentions toward prenatal breastfeeding; specifically, those who were positive expressed a greater intention to breastfeed than those who were pessimistic. However, it is crucial to frame our conclusions as associations rather than causal relationships due to the cross-sectional nature of this study.

According to the current study’s findings, age is a predictor of exclusive breastfeeding intentions. Our analysis revealed that older mothers (aged 30 and above) exhibited higher breastfeeding intentions compared to younger women aged 20–25. Factors such as maturity and experience may contribute to this trend, [10]. This aligns with findings from Dukuzumuremyi et al. [15], which state that older expecting mothers tend to have more positive intentions towards exclusive breastfeeding.

Additionally, the current study found that women in postgraduate education had significantly better breastfeeding intentions than those with secondary education (AOR 6.187). This can be attributed to higher educational attainment, which often correlates with greater health literacy and awareness regarding exclusive breastfeeding. Recent research supports the notion that maternal education is a reliable predictor of successful breastfeeding outcomes, emphasizing the importance of counseling and support throughout the prenatal and postnatal periods [29].

Another vital area of research is the connection between pregnancy planning and intentions for exclusive breastfeeding. Our findings show that women who planned their pregnancies had stronger intentions to breastfeed exclusively compared to those who had unplanned pregnancies. This can be explained by the fact that women who consciously choose to become pregnant often have more time to emotionally and psychologically prepare for motherhood and learn about breastfeeding. In contrast, women who experience unexpected pregnancies may face psychological stress or feelings of isolation, which can impact their breastfeeding intentions. Additionally, they may be less likely to participate in supportive community activities that promote health and well-being, such as programs promoting exclusive breastfeeding (EBF). This outcome aligns with the research by Ibrahim et al. [14], which found that being prepared for pregnancy significantly predicted a higher intention to practice exclusive breastfeeding (EBF) and underscores how the psychological readiness that comes with planned pregnancies can positively influence breastfeeding practices.

## 5. Strengths and Limitations

This study presents a central paradigm of behavior modification regarding exclusive breastfeeding (EBF). The sample size was calculated to achieve a 95% confidence interval, employing a systematic random sampling technique to recruit participants across two distinct regions in Saudi Arabia, thereby enhancing the generalizability of the findings. The data collection instruments utilized in this research were standardized and previously validated in methodological studies, contributing to the reliability of the data. Additionally, adjusted odds ratios were employed to accurately evaluate the influence of various predictors on maternal intention toward EBF.

Despite these strengths, several limitations need to be addressed more robustly. First, the study primarily relied on self-reported intentions regarding breastfeeding rather than actual breastfeeding behaviors. This reliance may not accurately reflect the participants’ subsequent actions postpartum. Second, the cross-sectional nature of the study restricts the ability to conclude temporal relationships between predictors and maternal intentions toward EBF. Without longitudinal data, it is challenging to ascertain which factors may precede or influence others. Third, the possibility of recall bias and social desirability bias should be emphasized more strongly. Participants may have overreported their intentions or provided responses they believed were more socially acceptable, which could distort the findings.

Lastly, the study did not assess actual breastfeeding practices or their determinants, as it focused solely on pregnant women and was not designed to be prospective. This limitation means that while maternal intentions were evaluated, the subsequent behaviors and experiences related to breastfeeding were not captured. Addressing these limitations in future research could enhance the understanding of the factors influencing EBF and its actual implementation in practice.

## 6. Conclusions

Although the current study’s findings indicate that over half of the primigravida women expressed a high intention to engage in exclusive breastfeeding (EBF), this outcome remains unsatisfactory. The bivariate analysis conducted in this study offers valuable insights into the factors influencing the breastfeeding attitudes of primigravida women in Saudi Arabia. Key factors identified include maternal age, educational level, ethnicity, pregnancy patterns, knowledge about breastfeeding, and overall attitudes toward exclusive breastfeeding. By pinpointing these modifiable predictors, healthcare providers can implement targeted interventions aimed at supporting and promoting exclusive breastfeeding among first-time mothers.

## 7. Recommendations

In light of the study’s findings, we recommend implementing comprehensive interventions that provide enhanced support during routine antenatal visits, especially for women facing unplanned pregnancies. Our strategies should focus on promoting exclusive breastfeeding through community-oriented initiatives and supportive policies, which focus on improving maternal and child health outcomes. Additionally, it is vital to create breastfeeding-friendly environments in workplaces and to advocate for policies that extend maternity leave and provide lactation breaks, thereby supporting breastfeeding practices effectively.

## Figures and Tables

**Table 1 healthcare-13-02938-t001:** Demographic characteristics of the studied participants (n = 436).

Variables	Categories	Frequency	%
Age in years	20–<25	121	27.8
	25–<30	234	53.7
	≥30 years old	81	18.5
Hospital	Hail	216	49.5
	Bisha	220	50.5
Education	Secondary	79	18.1
	University	281	64.4
	Postgraduate	76	17.4
Job	No	343	78.7
	Yes	93	21.3
Income	Low	47	10.8
	Moderate	332	76.1
	High	57	13.1
Nationality	Saudi	321	73.6
	Non-Saudi	115	26.4

**Table 2 healthcare-13-02938-t002:** Distribution of obstetric history among the studied participants (n = 436).

Variables	Categories	Frequency	%
Fetus	Single	357	81.9
	Twins	79	18.1
Abortion	No	307	70.4
	Yes	129	29.6
Complication	No	379	86.9
	Yes	57	13.1
Follow up	No	294	67.4
	Yes	142	32.6
Planned pregnancy	Yes	280	64.2
	No	156	35.8

**Table 3 healthcare-13-02938-t003:** Percentage and mean score of total knowledge, attitude, and intention regarding breastfeeding among studied participants (n = 436).

Variables	Categories	Frequency	%
Total knowledge	Inadequate	199	45.6
	Adequate	237	54.4
	MD ± SD (12.44 ± 5.90)		
Total attitude score	Negative attitude	202	46.3
	Positive attitude	234	53.7
	MD ± SD (58.77 ± 5.09)		
Intention	Low intention	182	41.7
	High intention	254	58.3
	MD ± SD (11.02 ± 2.38)		

**Table 4 healthcare-13-02938-t004:** Multivariable Logistic Regression of Factors Associated with High Exclusive Breastfeeding Intention (n = 436).

Variables	Categories	Low Intention		High Intention		AOR	Sig.	95% CI for (AOR)	
		No	%	No	%			Lower	Upper
Age in years	20–<25	80	(66.1%)	41	(33.9%)	Ref	0.40		
	25–<30	91	(38.9%)	143	(61.1%)	1.172	0.651	0.590	2.328
	≥30 years old	11	(13.6%)	70	(86.4%)	3.260	0.017	1.236	8.600
Hospital	Hail	117	(54.2%)	99	(45.8%)	Ref			
	Bisha	65	(29.5%)	155	(70.5%)	1.589	0.135	0.866	2.917
Education	Secondary	5	(71.4%)	2	(28.6%)	Ref	0.026		
	University	116	(46.8%)	132	(53.2%)	2.457	0.100	0.841	7.183
	Postgraduate	61	(33.7%)	120	(66.3%)	6.187	0.008	1.625	23.556
Job	Yes	124	(36.2%)	219	(63.8%)	Ref			
	No	58	(62.4%)	35	(37.6%)	0.786	0.569	0.343	1.799
Income	Low	15	(31.9%)	32	(68.1%)	Ref	0.035		
	Moderate	115	(34.6%)	217	(65.4%)	0.447	0.118	0.163	1.225
	High	52	(91.2%)	5	(8.8%)	0.156	0.010	0.038	0.636
Nationality	Saudi	158	(49.2%)	163	(50.8%)	Ref			
	Non-Saudi	24	(20.9%)	91	(79.1%)	2.470	0.014	1.200	5.085
Type of fetus	Single	123	(34.5%)	234	(65.5%)	Ref			
	Twins	59	(74.7%)	20	(25.3%)	0.541	0.182	0.220	1.332
Abortion	No	126	(41.0%)	181	(59.0%)	Ref			
	Yes	56	(43.4%)	73	(56.6%)	0.929	0.828	0.476	1.811
Complication	No	157	(41.4%)	222	(58.6%)	Ref			
	Yes	25	(43.9%)	32	(56.1%)	0.961	0.930	0.399	2.314
Follow up	No	86	(29.3%)	208	(70.7%)	Ref			
	Yes	96	(67.6%)	46	(32.4%)	0.521	0.077	0.253	1.073
Planned pregnancy	Yes	66	(23.6%)	214	(76.4%)	Ref			
	No	116	(74.4%)	40	(25.6%)	0.410	0.005	0.220	0.763
Total knowledge	Inadequate	135	(67.8%)	64	(32.2%)	Ref			
	Adequate	47	(19.8%)	190	(80.2%)	0.402	0.002	0.226	0.717
Total attitude score	Negative attitude	111	(55.0%)	91	(45.0%)	Ref			
	Positive attitude	71	(30.3%)	163	(69.7%)	0.396	0.002	0.223	0.705

AOR = Adjusted Odds Ratio; CI = 95% Confidence Interval; Ref = Reference category; *p* < 0.05; Dependent variable: high exclusive breastfeeding intention; predictors: age, education, nationality, planned pregnancy, knowledge, and attitude.

## Data Availability

The datasets used or analyzed in this study are available from the corresponding author upon request. The data are not publicly available due to privacy concerns, and ethical restrictions.

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
