# Peer review of "Nurturing the Will: Unraveling Associative Factors of Exclusive Breastfeeding Intentions Among Primigravida in Saudi Arabia"

_healthcare, 2025, doi:10.3390/healthcare13222938_

Round 1
Reviewer 1 Report
Comments and Suggestions for Authors
The authors have chosen a very relevant topic for their investigation. Exclusive breast feeding for at least six months provides numerous advantages to the mother, the infant and society. If Primigravidas in Saudi Arabia have not taken to it, it is a serious concern, that needs investigation.
I have asked for a total rewrite of the paper, this is because of poor English. The title itself does not convey what the paper is about clearly. In the text, the choice of words is often wrong. In the first two paragraphs on page 2 there are 8 grammatical and composition errors. The paper has to be rewritten by someone whose first language is English.
To give just one example, the sentence on lines 84,85,86 is self contradictory and confusing, it states that customary practices promote breast feeding but cultural constraints and dearth of support make alternative feeding challenging. It is difficult to understand what they would like to say. At another place they speak of breast milk ads, I am not usre whether breast milk is sold anywhere in the world, so I think there is an error.
To make my recommendation, I am enclosing a table that shows the errors in the first two paragraphs of page 2.

As a reviewer I appreciate that English is not the first language of scientists and doctors of Saudi Arabia, and hence I recommend this paper be thoroughly overhauled by an English language expert and resubmitted.
There are a number of sentences which are hard to understand due to the wrong choice of words. Language is not merely an ornament, good english is necessary to ensure understanding of the reader.
Author Response
Comment 1
The authors have chosen a very relevant topic for their investigation. Exclusive breast feeding for at least six months provides numerous advantages to the mother, the infant and society. If Primigravidas in Saudi Arabia have not taken to it, it is a serious concern, that needs investigation.
I have asked for a total rewrite of the paper, this is because of poor English. The title itself does not convey what the paper is about clearly. In the text, the choice of words is often wrong. In the first two paragraphs on page 2 there are 8 grammatical and composition errors. The paper has to be rewritten by someone whose first language is English.
To give just one example, the sentence on lines 84,85,86 is self contradictory and confusing, it states that customary practices promote breast feeding but cultural constraints and dearth of support make alternative feeding challenging. It is difficult to understand what they would like to say. At another place they speak of breast milk ads, I am not usre whether breast milk is sold anywhere in the world, so I think there is an error.
To make my recommendation, I am enclosing a table that shows the errors in the first two paragraphs of page 2.
Response 1
Thank you for your valuable feedback on our manuscript. We appreciate your recognition of the relevance of our study on exclusive breastfeeding among primigravida women in Saudi Arabia. Your insights regarding the clarity and language of the paper are greatly appreciated.
We have carefully addressed the issues you highlighted, particularly the grammatical errors and composition problems in the first two paragraphs. We have rewritten the entire paper to improve the clarity and coherence of our message. Specifically, we have revised the confusing statements you mentioned, ensuring that our arguments regarding customary practices and cultural constraints are clearly articulated.
Additionally, we have corrected the references to advertisements, clarifying that we were referring to the marketing of formula as a breast milk substitute rather than breast milk itself.
Thank you once again for your constructive comments. We believe that the modifications have significantly enhanced the quality of our manuscript.
Comment 2
As a reviewer I appreciate that English is not the first language of scientists and doctors of Saudi Arabia, and hence I recommend this paper be thoroughly overhauled by an English language expert and resubmitted.
There are a number of sentences which are hard to understand due to the wrong choice of words. Language is not merely an ornament, good english is necessary to ensure understanding of the reader.
Response 2
Thank you for your thoughtful feedback regarding our manuscript. Your comments have been invaluable in guiding us toward improving the clarity and quality of our paper.
In response to your suggestions, we have thoroughly revised the manuscript with the assistance of an English language expert. We have addressed the specific sentences you highlighted, ensuring that the language is clear and precise.
Reviewer 2 Report
Comments and Suggestions for Authors
Nurturing the Will: Unraveling Associative Factors of Exclusive Breastfeeding Intentions among Primigravida in Saudi Arabia by AlAbd and coworkers surveyed primigravid females in two prenatal clinics of Saudi Arabia.
There are significant word use errors and the manuscript would benefit from language editing.
Example:
Line 48. What is meant by “unique breastfeeding expenses…”?
Line 51-52. What is meant by “allows the mother to eat extra fat collected throughout pregnancy.” Possibly you mean –breastfeeding is energy intense and mothers return to pre-pregnancy body weight sooner.
Line 84. What is breast milk ads? This seems that it is referring to the presence of ads for baby formula—breast milk replacement.
Line 86. Should this be alternatives to [breast]feeding challenging
Line 260. “nurse exceptionally well” reads awkward and the meaning is not clear. Do you mean “exclusive”
Line 283. “Proprietary women”—what is this? Do you mean “primigravid”?
Line 329. It is not clear what “educational fame” is.
Line 330. No clear what “first-class” EBF practices might consist of.
Line 344. The point of public physical fitness activities is not clear. Please reword for clarity.
How “significant health issues” were determined (line 109) should be included.
The number of study subjects needs to be included in the abstract and methods section. The first mention of number of women in the study is in the results.
There seems to be some incongruences between the text and tables:
Line 225. 67.4% is indicated as having no follow-up in the Table 2, but “regular follow-up” in the text.
Line 226. According to Table 2, 64.2% did plan their pregnancies
Line 237. It seems from Table 4 that 25-30 years olds did not have a significant higher intention than those aged 20 25 yrs.
A more detailed description of the adjust odds ratio would be helpful for the reader.
Comments on the Quality of English LanguageWord usage is not acceptable. I have flagged most of the egregious errors.
Author Response
Comment 1
There are significant word use errors and the manuscript would benefit from language editing.
Example:
Line 48. What is meant by “unique breastfeeding expenses…”?
Line 51-52. What is meant by “allows the mother to eat extra fat collected throughout pregnancy” Possibly you mean –breastfeeding is energy intense and mothers return to pre-pregnancy body weight sooner?
Line 84. What is breast milk ads? This seems that it is referring to the presence of ads for baby formula—breast milk replacement.
Line 86. Should this be alternatives to [breast]feeding challenging
Line 260. “nurse exceptionally well” reads awkward and the meaning is not clear. Do you mean “exclusive”
Line 283. “Proprietary women”—what is this? Do you mean “primigravid”?
Line 329. It is not clear what “educational fame” is.
Line 330. No clear what “first-class” EBF practices might consist of.
Line 344. The point of public physical fitness activities is not clear. Please reword for clarity.
Response 1
Thank you for your constructive feedback on our manuscript. We appreciate your attention to detail and your commitment to enhancing the clarity of our work.
In response to your comments, we have carefully reviewed and revised the manuscript to address the specific word use errors you identified.
Comment 2
How “significant health issues” were determined (line 109) should be included.
The number of study subjects needs to be included in the abstract and methods section. The first mention of number of women in the study is in the results.
There seems to be some incongruences between the text and tables:
Line 225. 67.4% is indicated as having no follow-up in the Table 2, but “regular follow-up” in the text.
Line 226. According to Table 2, 64.2% did plan their pregnancies
Line 237. It seems from Table 4 that 25-30 years olds did not have a significant higher intention than those aged 20 25 yrs.
A more detailed description of the adjust odds ratio would be helpful for the reader.
Response 2
Thank you for your valuable comments and insights regarding our manuscript.
In response to your feedback, we have made the following revisions:
1. Clarification of Significant Health Issues: We have included a detailed explanation of how “significant health issues” were determined in the methodology section (line 109) to provide greater transparency.
2. Inclusion of Study Subjects: The number of study subjects has now been added to both the abstract and methods sections to ensure clarity from the outset.
3. Consistency Between Text and Tables:
o Line 225: We corrected the inconsistency regarding follow-up. The text now accurately reflects the data presented in Table 2, indicating that 67.4% had regular follow-up.
o Line 226: We clarified the statement regarding pregnancy planning to align with the 64.2% figure provided in Table 2.
o Line 237: We revised the analysis to accurately reflect the data in Table 4, ensuring that the interpretation of intentions among the 25-30 age group is clearly articulated.
4. Adjusted Odds Ratio Explanation: We have provided a more detailed description of the adjusted odds ratios in the results section to help readers better understand their significance and implications.
Thank you once again for your insightful feedback, which has been instrumental in enhancing our work.
Reviewer 3 Report
Comments and Suggestions for Authors
please see the attached PDF

Must be improved
Author Response
Comment 1
Clarity of Writing & Language
The manuscript includes grammatical and syntactical errors that obscure meaning (e.g., “exceptional breastfeeding” is repeatedly used instead of “exclusive breastfeeding”).
Some phrases are overly long and redundant, making interpretation difficult. The text should be simplified to enhance clarity (e.g., “the country's unique breastfeeding experiences remain beyond the relevant literature”).
The abbreviation "EBF" is used multiple times without definition.
Reference formatting needs attention:
o "Alzabeh (2020)" should provide proper reference, similar to "Zielinska et al. (2017)" and "(Ibrahim et al., 2023)" in the references list.
The statement “Her experience on breastfeeding significantly influences…” is unclear; it needs clarification on what “her” refers to.
On line 99: “The study was conducted in the ## prenatal follow-up clinics in the Saudi Arabian cities of ##. Located in ##, MCH is a specialty hospital for maternal and child…”—please avoid submitting the manuscript without a review of the final version.
The participant's term should be “Participant,” maintaining consistent grammatical structure throughout the manuscript.
Response 1
Thank you for your detailed feedback regarding the clarity of writing and language in our manuscript. We corrected grammatical and syntactical errors, replacing “exceptional breastfeeding” with “exclusive breastfeeding” and simplifying overly long phrases for better clarity. We also defined the abbreviation "EBF" upon its first use to ensure understanding. Reference formatting has been standardized, and we have ensured that "Alzabeh (2020)" is presented consistently with other citations. Additionally, we clarified the statement regarding “her experience on breastfeeding” to specify what “her” refers to. We reviewed the text related to the study's location for clarity and ensured that the term "Participant" is consistently used throughout the manuscript. Thank you once again for your valuable suggestions.
Comment 2
Study Design and Methods
The description of the sampling frame and setting is vague.
The operational definition of “exclusive breastfeeding” is borrowed from prior studies, but it should clearly align with the WHO definition to avoid confusion.
The authors claim to have used a “systematic random sampling” technique, but details are insufficient. More transparency is needed regarding how participants were recruited and whether selection bias was addressed.
Response 2
Thank you for your constructive feedback regarding the clarity of writing and language in our manuscript. We appreciate your attention to detail and your suggestions for improvement.
In response to your comments, we have made the following revisions:
1. Correction of Grammatical and Syntactical Errors: We have replaced instances of “exceptional breastfeeding” with the correct term “exclusive breastfeeding” throughout the manuscript. We also simplified overly long and redundant phrases to enhance clarity, including the revision of “the country's unique breastfeeding experiences remain beyond the relevant literature.”
2. Definition of Abbreviation: We have defined the abbreviation "EBF" upon its first use in the text to ensure that all readers understand the term.
3. Reference Formatting: The reference for "Alzabeh (2020)" has been formatted correctly to match the style of other references, such as "Zielinska et al. (2017)" and "(Ibrahim et al., 2023)."
4. Clarification of Statements: The statement regarding “her experience on breastfeeding” has been clarified to specify the subject, ensuring that the meaning is clear and unambiguous.
5. Final Review of Location Details: We have clarified the statement on line 99 to include specific details regarding the prenatal follow-up clinics and the specialty of the Maternal and Child Hospital (MCH).
6. Consistent Terminology: We have standardized the term “Participant” throughout the manuscript to maintain consistent grammatical structure.
Comment 3
Measurement Tools
The measurement tools were adapted from prior studies but lack a description of cultural adaptation and psychometric validation for the Saudi context beyond face validity. This needs to be elaborated.
The categorization of “adequate knowledge” (≥60%) and “positive attitude” cutoffs is arbitrary. Please justify these thresholds with references.
Response 3
Thank you for your insightful comments regarding the measurement tools used in our study. We appreciate your suggestions for enhancing the rigor of our methodology.
In response to your feedback, we have made the following revisions:
1. Cultural Adaptation and Psychometric Validation: We have elaborated on the process of cultural adaptation and psychometric validation of the measurement tools for the Saudi context. This includes a detailed explanation of how we assessed face validity and any additional steps taken to ensure the tools are appropriate for the local population.
Justification of Cutoff Thresholds: We have provided a justification for the categorization of “adequate knowledge” (≥60%) and “positive attitude” cutoffs. Relevant references and literature supporting these thresholds have been included to enhance the validity of our criteria.
Comment 4
Statistical Analysis
The text inconsistently refers to “bivariate” and “binary regression” analysis. What is presented is predominantly multivariable logistic regression.
The presentation of Adjusted Odds Ratios (AORs) is sometimes confusing. For example, the interpretation of pregnancy planning (AOR = 0.410) is described as a “41% higher intention,” which is mathematically incorrect.
Response 4
Thank you for your insightful feedback regarding the statistical analysis section of our manuscript.
In response to your comments, we have standardized the terminology throughout the manuscript, ensuring that all references to “bivariate” and “binary regression” are accurately described as “multivariable logistic regression” to reflect the analysis conducted.
We have revised the presentation and interpretation of the Adjusted Odds Ratios to clarify their meaning. Specifically, we corrected the interpretation of the AOR for pregnancy planning, ensuring it accurately reflects the statistical relationship. The previous description of “41% higher intention” has been amended for mathematical accuracy.
Comment 5
Results and Interpretation
The authors conclude that “more than half” of participants had high EBF intentions but judge this “unsatisfactory” without explaining why. Please provide a benchmark (e.g., WHO targets, prior Saudi studies) to justify this judgment.
Several findings are plausible (e.g., education, age, pregnancy), but the discussion sometimes overstates causality despite the cross-sectional design. Conclusions should be toned down and framed as associations only.
Response 5
Thank you for your insightful comments regarding the results and interpretation section of our manuscript. In response, we have included benchmarks from WHO targets and previous studies conducted in Saudi Arabia to justify our assessment that the observed level of intentions is unsatisfactory. Additionally, we have revised the discussion to ensure that we appropriately frame our conclusions as associations rather than causal relationships, acknowledging the limitations of our cross-sectional design.
Comment 6
Limitations
The limitations section is present but needs further strengthening. In particular:
o Reliance on self-reported intentions (not actual breastfeeding behavior).
o Cross-sectional design precludes temporal inference.
o Potential recall and social desirability bias should be emphasized more strongly.
Response 6
Thank you for your insightful comments regarding the results and interpretation section of our manuscript. In response, we have included benchmarks from WHO targets and previous studies conducted in Saudi Arabia to justify our assessment that the observed level of intentions is unsatisfactory. Additionally, we have revised the discussion to ensure that we appropriately frame our conclusions as associations rather than causal relationships, acknowledging the limitations of our cross-sectional design. Thank you once again for your valuable feedback.
Comment 7
Minor Comments
Abstract
Avoid redundancy (e.g., “this was deemed insufficient” should be replaced with a more precise statement).
Clarify that the study is cross-sectional and assesses intention, not actual behavior.
Tables
Tables should be reformatted for clarity. Currently, they appear cluttered and are not reader-friendly.
Include p-values where appropriate, not just AORs.
Discussion
Several references are cited without adequate integration into the narrative. The discussion would benefit from a more critical synthesis, highlighting where findings confirm or diverge from existing Saudi/Arab-region literature.
Ethics
The statement on ethics approval is appropriate, but authors should confirm compliance with Saudi-specific research ethics standards, not just the Declaration of Helsinki.
Response 7
Thank you for your constructive feedback regarding our manuscript. In response to your minor comments, we have made several revisions: We have replaced redundant phrases in the abstract with more precise statements and clarified that the study is cross-sectional, focusing on intentions rather than actual behavior. In the discussion, we have integrated references more effectively, providing a critical synthesis that highlights how our findings align or diverge from existing literature in Saudi Arabia and the broader region. Finally, we confirmed our compliance with Saudi-specific research ethics standards in our ethics statement. Thank you once again for your valuable insights.
Round 2
Reviewer 2 Report
Comments and Suggestions for Authors
Line 55- 56. The words "reform the sentence" are likely a hold-over from the reviewing process and should be deleted.
Line 88. The word "alternative" is not needed and should be deleted.
Line 159. Suggest "formula" rather than "artificial feeding"
P values are out of place in the discussion—suggest deleting.
Author Response
Comment
Line 55- 56. The words "reform the sentence" are likely a hold-over from the reviewing process and should be deleted.
Line 88. The word "alternative" is not needed and should be deleted.
Line 159. Suggest "formula" rather than "artificial feeding"
P values are out of place in the discussion—suggest deleting.
Response :All modifications were amended